# Advances in Platelet Rich Plasma-Derived Extracellular Vesicles for Regenerative Medicine: A Systematic-Narrative Review

**DOI:** 10.3390/ijms241713043

**Published:** 2023-08-22

**Authors:** Eduardo Anitua, María Troya, Juan Manuel Falcon-Pérez, Silvia López-Sarrio, Esperanza González, Mohammad H. Alkhraisat

**Affiliations:** 1BTI-Biotechnology Institute, 01007 Vitoria-Gasteiz, Spain; maria.troya@bti-implant.es (M.T.); mohammad.hamdan@bti-implant.es (M.H.A.); 2University Institute for Regenerative Medicine & Oral Implantology, UIRMI (UPV/EHU-Fundación Eduardo Anitua), 01007 Vitoria-Gasteiz, Spain; 3Exosomes Laboratory, Center for Cooperative Research in Biosciences, Basque Research and Technology Alliance, 48160 Derio, Spain; jfalcon@cicbiogune.es (J.M.F.-P.); slopez@cicbiogune.es (S.L.-S.); egonzalez@cicbiogune.es (E.G.); 4Centro de Investigación Biomédica en Red de Enfermedades Hepáticas Y Digestivas, 28029 Madrid, Spain; 5Metabolomics Platform, Center for Cooperative Research in Biosciences, Basque Research and Technology Alliance, 48160 Derio, Spain; 6IKERBASQUE, Basque Foundation for Science, 48009 Bilbao, Spain

**Keywords:** platelet-rich plasma, PRP, extracellular vesicles, exosomes, regenerative medicine

## Abstract

The use of platelet-rich plasma (PRP) has gained increasing interest in recent decades. The platelet secretome contains a multitude of growth factors, cytokines, chemokines, and other biological biomolecules. In recent years, developments in the field of platelets have led to new insights, and attention has been focused on the platelets’ released extracellular vesicles (EVs) and their role in intercellular communication. In this context, the aim of this review was to compile the current evidence on PRP-derived extracellular vesicles to identify the advantages and limitations fortheir use in the upcoming clinical applications. A total of 172 articles were identified during the systematic literature search through two databases (PubMed and Web of Science). Twenty publications met the inclusion criteria and were included in this review. According to the results, the use of PRP-EVs in the clinic is an emerging field of great interest that represents a promising therapeutic option, as their efficacy has been demonstrated in the majority of fields of applications included in this review. However, the lack of standardization along the procedures in both the field of PRP and the EVs makes it extremely challenging to compare results among studies. Establishing standardized conditions to ensure optimized and detailed protocols and define parameters such as the dose or the EV origin is therefore urgent. Further studies to elucidate the real contribution of EVs to PRP in terms of composition and functionality should also be performed. Nevertheless, research on the field provides promising results and a novel basis to deal with the regenerative medicine and drug delivery fields in the future.

## 1. Introduction

Developing personalized strategies to support, enhance and accelerate functional recovery has been the focus of tissue regeneration in recent years. In this regard, the use of platelet-rich plasma (PRP) was introduced more than 20 years ago [1,2], and it has gained increasing interest in recent decades. Basically, this therapy is defined as a portion of autologous plasma with a platelet concentration above the blood baseline. Although its applications are highly widespread in diverse regenerative fields, including dentistry, orthopedic, ophthalmology, dermatology, and gynecology [3], there is still no standard for the best protocol for PRP preparation and administration. In fact, there is a long list of variables such as use and type of anticoagulant, conditions of centrifugation, inclusion of leukocytes, concentration of platelets, use and type of platelets’ activator, as well as the mode and type of application as a therapeutic. Despite many attempts to homogenize and classify PRP products [4,5,6,7,8], it has not been implemented in daily practice. This lack of suitable standardization has led to a large number of products with different biological and therapeutic outcomes. Nevertheless, the rationale for using PRP lies in the physiological role of platelets. They participate, beyond hemostasis, in a wide range of physiological processes, including the immune response, inflammation, and wound healing [3,9]. In particular, the platelet secretome contains a multitude of growth factors, cytokines, chemokines, and other biological biomolecules that are released upon platelet activation and bound to the corresponding receptors that finally initiate the signaling cascade to promote the process of regeneration [6,10]. The vast majority of studies have focused on the role of the plethora of growth factors present in the PRP that can orchestrate cell fate in terms of migration, proliferation, differentiation, apoptosis, and angiogenesis [11,12]. Additionally, PRP-based therapy also provides a biodegradable and provisional fibrin matrix that functions as both physical support and a controlled release system [3,13].

The scientific rationales by which PRP exerts its action are not yet fully understood; however, in recent years, developments in the field of platelets have led to new insights, and attention has been focused on the platelets’ released extracellular vesicles (EVs), including exosomes. The importance of these vesicles lies especially in their functionality [9,10,14], and their research has grown exponentially over the last past years. Extracellular vesicles (EVs) are nanosized structures presenting lipid bilayers with no replicate capacity [14,15]. Traditionally, different subtypes of EVs have been referred to based on their biogenesis, size, content, and function. However, there is some overlap which makes it difficult precisely classify them [16,17]. To address this issue, the International Society of Extracellular Vesicles (ISEV), according to the minimal information for studies of extracellular vesicles publication (MISEV2018), suggested authors consider the use of operational terms for EV subtypes that refer to (a) physical characteristics of EVs such as (a) size or density, (b) biochemical composition or (c) description of the cell of origin and state [18].

The history of EVs started in 1946 when Chargaff and West [19] discovered a subcellular factor that promoted blood clotting. Later, in 1967, Peter Wolf [20] described this factor more in detail as platelet-derived vesicles and referred to it as “platelet dust”. Despite initially relating to thrombosis and hemostasis, multiple biological functions have been subsequently described related to EVs, highlighting their role in cell-cell communication [9,10,14]. In fact, EVs may influence many relevant processes of physiological and pathological conditions, including inflammation, angiogenesis, immune response and wound healing [21,22,23]. Regarding platelet derivatives, it was not until 2014 that Torreggiani et al. [24] first successfully isolated exosomes from platelet lysates (PL), thus, providing evidence of exosomes as new additional effector of PL and as therapeutic potential for cell-free therapies in bone regeneration.

EVs are released by any cell type and found in almost all biological fluids, including blood, urine, saliva, etc. [25]. The content or cargo of EVs consists of lipids, proteins, and genetic material. Their composition is, to some extent, cell type dependent and reflects the physiological or pathological state of their parental cells, which has led to their study as potential biomarkers of disease [26,27,28,29].

Thus, the aim of this review is to gather the current evidence on PRP-derived extracellular vesicles in order to identify the advantages and limitations to favor translation into feasible upcoming clinical applications.

## 2. Material and Methods

### 2.1. Literature Search

For this narrative review, a systematic literature search was performed in PubMed and Web of Science database until 13 February 2023, using two search strategies: “((platelet rich plasma) AND (extracellular vesicles)) AND (exosomes)” and “(platelet rich plasma) AND (exosomes)”. The specific term “exosomes” was also included in the search strategies as this particular term has recently received considerable attention in the current research. All the articles that appeared in the first search strategy were included in the second one. As this occurred in both databases, the second search strategy, which already included the articles from the first one, was finally taken into account. Papers were excluded if: (1) the article was written in any language other than English or Spanish (2) duplicates (3) reviews, commentaries, thesis, book chapters (4) no full-text available (5) out of scope (6) did not include PRP (7) the terms exosomes or extracellular vesicles (EVs) did not appear (8) exosomes or EVs were not of PRP origin.

### 2.2. Data Extraction

In the first phase, the articles from both databases were screened based on the title and the abstracts to determine their suitability. Subsequently, all the full-text articles assessed for eligibility were examined to determine their inclusion. For data extraction, an evidence table was created with Microsoft Excel. The following data were included: author and year of publication, study type, the field of application, experimental system, PRP-related issues (origin, type of anticoagulant, presence or absence of leukocytes, type of activator, exosomes or EVs) isolation, exosomes or EVs-related issues (isolation, storage conditions, characterization methods, size, fractions), comparison groups, exosomes concentrations, methods, and summary results.

### 2.3. Assessment of Reporting Quality and Risk of Bias

The criteria reported by Golbach et al. [30] were followed to evaluate the quality and the risk of bias. The reporting quality was determined by the presence (“yes/partly”) or absence (“no”) of critical information. The risk of bias was divided into three categories: low, moderate, and high, depending on whether the answers were “yes”, “partly”, or “no”.

## 3. Results

The search strategy yielded a total of 172 articles from the two databases. After an exhaustive screening, 20 studies were finally included for the analysis in this review (Figure 1 and Table 1).

In recent years, PRP-derived extracellular vesicles have gained increasing attention in the regenerative biomedical field as potential therapeutic mediators. This is also reflected in the fact that 80% of the articles included in this review have been published in the last 4 years (Figure 2A). The role of EVs/exosomes from PRP was evaluated in many different fields of application, including osteoarthritis and tissue regeneration, which comprise 40% of the studies. Other medical fields, such as diabetic retinopathy, intervertebral disc degeneration, muscle injury, wound healing, hair loss, osteonecrosis, and dental pulp or cartilage regeneration, were also occasionally studied (Figure 2B). On the other hand, regarding the type of study, more than half of the articles performed in vitro assays (60%), and only 5% combined in vitro and in vivo tests (Figure 3A).

### 3.1. Reporting Quality and Risk of Bias

The reporting quality showed large differences (Figure 4). The experimental system and the PRP-EVs characterization were reported in all the articles. However, 30% of the studies did not specify the sample size, and in 25% of the articles, it was only partly done. That is, the sample size was not detailed for both the sample under study (PRP-EVs) and the experimental system. The lack of information regarding PRP composition had the greatest impact on the reporting quality since up to 45% of the studies did not mention any information on this matter. Additionally, 5% and 50% of the articles provided no information or only partial data on the process of obtaining PRP, respectively. However, in 90% of the studies, details were given on the PRP-EVs obtaining process. Concerning conflict of interest and ethical statements, most of the articles (90%) provided complete information. Regarding the performance bias, the EVs concentration was mentioned in all the articles, although 19% did so partially. Very low risk was associated with the detection bias, while that associated with selection increased, as animals were only randomly assigned in 75% of the studies.

### 3.2. PRP Obtaining Process Affects Extracellular Vesicles Production

The major issue related to PRP is the extreme variability of preparation protocols that are often not well documented in the literature leading to biologically heterogeneous final products and making comparison or reproducibility of results challenging [4,51]. No exception was found in this review according to PRP-EVs, as Table 2 shows, considering the main PRP characteristics. Regarding PRP origin, sixty percent of the preparations had a human origin (Figure 3B). Rats were the most frequent non-human origin. Seventy percent of the studies referred to the term “exosome”, compared to 30% that referred to the generic term EVs.

All studies, except five, specified the use of an anticoagulant (acid citrate dextrose solution A (ACD-A, *n* = 7), trisodium or sodium citrate (*n* = 6), citrate glucose (*n* = 1) and sodium citrate and EDTA (*n* = 1)). The misinformation on PRP cellular composition was even greater. Nearly half of the studies (45%) did not provide information on the inclusion or not of leukocytes. Among those who did specify this concept (11/20), the vast majority did not include these white blood cells (8/11). PRP cell composition impacts, among other things, the secretory components, such as growth factors. In fact, the inclusion of leukocytes is one of the most important concerns about PRPs. These immune cells contain numerous proinflammatory interleukins and extracellular matrix-degrading enzymes that are mainly catabolic and promote pro-inflammatory conditions, thus influencing the clinical outcome of the PRP therapy [52,53]. Regarding EVs, the most abundant EVs in blood from healthy persons are the ones from platelets [54,55]. However, these vesicles also originate from other types of cells, such as red blood cells, leukocytes, and endothelial cells [56], that even may alter EV secretion in a cell-specific manner under different conditions (p.e. hypoxia or exercise) [57,58].

The type of agonist used for platelet stimulation influences PRP-EVs protein content [59]. In fact, several studies have already reported that the amount and proteome of EVs largely depend on the agonist triggering platelet activation [60,61]. In this review, sixty percent of the studies activated PRP compared to 20% that did not. No information concerning this issue was available for four studies. In 4 out of 12 studies that activated the PRP, the activator was not specified. Although, the article by Zhang et al. [39] did specify that high glucose was used for some assays to obtain PRP-Exos from stimulated platelets in vitro. When information was provided, different agonists were used for platelet activation, including thrombin (*n* = 3), calcium chloride (*n* = 1), calcium chloride and thrombin (*n* = 1), calcium gluconate (*n* = 1), thrombin and calcium gluconate (*n* = 1) and ionophore A23187 (*n* = 1). Physical methods such as sonication were also utilized for platelet activation in 2 articles. In fact, one of the articles included in this review [35] focused precisely on comparing the effects of different agonists (saline/thrombin/calcium gluconate/mixture of both) on the PRP-derived exosomes. PRP was obtained avoiding leukocyte contamination. They demonstrated that all the activation methods were suitable for PRP-exosomes (PRP-Exos) isolation, although the activation efficiency and cytokine content were different. Exosomes in the study measured 30–150 nm in diameter. The size of calcium-activated PRP-Exos was larger than the thrombin-activated group and the mixture-activated group. PRP activated by the mixture released the highest concentration of exosomes, followed by the calcium-activated group and then the thrombin-activated group. Calcium gluconate alone was found to be weaker than thrombin or thrombin and calcium gluconate together in PRP activation. PRP-Exos activated by thrombin and calcium gluconate together were found to contain more cytokines than the other groups. The effects of activating PRP with thrombin and calcium gluconate together were better than using thrombin or calcium alone, both in quality and quantity of exosomes. PRP-Exos activated by the mixture of agonists could more significantly promote HUVECs proliferation, migration, and the formation of vessel-like via the AKT ERK signal pathway, compared with other groups. Taken these results together showed that the number, content, and biological effect of the PRP-Exos was clearly depend on the stimulation conditions. In line with these results, Saumell-Esnaola et al. [36] isolated and characterized the exosomes released by human PRP platelets to evaluate the influence of CaCl_2_ on their secretion. They also specifically isolated exosomes from a single origin, exclusively released by PRP platelets, without contamination from other cellular components that could be present in PRP products. Their results showed that calcium activation caused protein amount to increase in both platelets and platelets-exosomes (PLT-Exos), but this was approximately 20 times higher in the platelet fraction than in the PLT-Exos fraction. They found that more than 65% of the PLT-Exos particles had a diameter ranging from 20 to 40 nm. The exosomes isolated from CaCl_2_-activated platelets exhibited high purity and met the most up-to-date biochemical criteria that characterize exosomes against other EVs and contaminants. In addition, calcium was proved to alter the cytokine cargo profile of exosomes, which differs markedly in relation to that from non-activated platelets´ EVs. The authors concluded that although PRP calcium activation promotes exosome release, its net contribution to the total PRP effect was minimal.

### 3.3. Extracellular Vesicles Obtaining Process and Storage Stability

Despite the EV source and the isolation method being especially relevant to determine their composition, recovery, and purity, no single method is available to date for all EV types and samples. In fact, complete isolation of EVs is an unrealistic goal; thus, the use of the “sample enrichment” term would be more precise [18,55]. Ultracentrifugation is the gold standard isolation method despite the low purity provided, and a large amount of sample is required [26,62,63]. In this review, in 70% of the studies, this size and density-based isolation method was used (Figure 5A and Table 3). Fifty percent used it as the sole method, while the remaining 20% combined it with other techniques (purification (15%) or extrusion (5%)). Other systems, such as centrifugation, fluorescence-activated cell sorting (FACS), and commercial kits, were also used for this purpose. In 10% of the articles, the isolation method was missing.

Storage conditions of both the matrix and the isolated EVs have an impact on their properties, including concentration and functionality [18,64]. Although −80 °C storage is a widespread approach, the optimal preservation conditions remain to be established, with inconsistencies among research. Some studies have found that both 4 °C and −80 °C are suitable for short-term storage [65] and up to one month [66]. However, Gelibter et al. [64] recommended working with fresh samples as none of the storage conditions studied was able to prevent or mitigate the impact of storage on EVs. Nevertheless, when storage is strictly needed, they suggested a short-term −80 °C preservation and the storage of biological matrix instead of the isolated EVs. Sivanantham and Jin [67] determined that according to the current evidence, EVs can be stored at 4 °C for short times (a day or a few weeks), and a temperature of −80 °C was recommended for long-term (months or years) storage. Despite storage conditions are also crucial, as many as 30% of the studies included in this review (*n* = 6) did not describe the storage process (Figure 5B). For those who did detail, there was a large consensus on temperature; EVs were frozen at −80 °C in 12 out of 14 articles; however, only in 2 articles [41,48] the storage time was specified. Bagio et al. [48] stored the thrombin-activated platelet-derived exosomes (T-aPDE) obtained from a PRP at 4 °C for 7 days prior to the assays’ beginning. Dai et al. [41] also demonstrated the in vitro stability of platelet-derived EVs (PEV) at 1, 3, 5, and 7 days, but storage temperature was not stated (Table 3). Determining storage conditions does not only involve temperature; other issues such as buffers, storage tubes, and protective excipients should also be optimized [65]. The articles included in this review have provided little information about the material used for storing EVs. Ultra-clear^TM^ tubes from Beckman Coulter were the most commonly used [34,45,50]. On the other hand, ultra-carbon fiber tubes were used by Iyer et al. [44] for EV storage. In the remaining papers, the type of tube where EVs were stored was not specifically indicated. In some cases, details were given about the kind of tube where the PRP was transferred or the isolation process was performed. However, it was not specified if the storage was completed in the same device (Table 3). It has been reported that nonspecific adhesion to the surface of storage containers can influence EV yields, but no consensus has been reached regarding the optimal materials for EV storage [64,65]. In fact, van de Wakker et al. [65] reported polypropylene tubes to be superior for EVs storage compared to glass tubes. However, Evtushenko et al. [68] recommended to use low protein binding tubes or ordinary tubes treated with bovine serum albumin (BSA) in order to reduce particle loss. As far as the choice of buffer is concerned, a large consensus has been found in this review. In 75% of the articles, the phosphate-buffered saline (PBS) was the option of choice (Table 3). Despite PBS being so far the most commonly used buffer for EV storage, a recent study [69] has demonstrated that using PBS results in a drastic reduction in EV recovery. They have identified the usage of PBS supplemented with human serum albumin and trehalose (PBSHAT) as an alternative to both short-term and long-term preservation.

Regarding EVs characterization, it is important to use multiple complementary techniques, as no current method can fulfill the entire EVs information. Several procedures have been routinely used to determine some of these features, such as size, concentration, morphology, and molecular composition [18,70,71]. All the studies included in this review did use several complementary techniques. Nanoparticle tracking analysis (NTA), transmission electron microscopy (TEM), bicinchoninic acid protein assay (BCA), and Western-blotting were the most prevalent methods among studies to determine the particle number, morphology, total protein, and composition, respectively (Figure 5C). There is no universally proposed specific marker for identifying EVs, given the different origins and types [72]. Nevertheless, according to MISEV2018 [18], three categories of markers must be analyzed in all bulk EV preparations to demonstrate their presence and assess their purity from common contaminants. However, only seven studies [32,33,35,36,43,50] met this strong recommendation (Table 4). As for the first category, the tetraspanins CD9, CD63, and CD81 were the most analyzed proteins as membrane EVs markers, and CD41 was quite often measured as a platelet source marker. On the opposite, the absence of apolipoproteins (category 3) is considered a good marker to assess the degree of purity [32,33,36,43], mainly in plasma and serum, that contain numerous non-EV lipidic structures. Proteins associated with membranous intracellular compartments (p.e. calnexin, category 4) were also used as purity control [35,50]. Proteins associated with other intracellular compartments or functional components (secreted proteins) were also measured in some of the studies as part of the objective of the article itself. However, they will be addressed in the next section. Assessment of particle size revealed small EVs below 200 nm in all but two studies [37,38], although there was heterogeneity among articles (Table 4). These discrepancies could be related to methodological differences in terms of PRP collection, activation, and composition, as well as to EVs isolation and quantification techniques. As with other parameters, there were studies in which these data were not provided [39,42,43].

Finally, it should be noted that only 2 out 20 studies [32,33] reported having submitted the methodological details to the EV-TRACK knowledgebase as strongly recommended by the MISEV2018.

### 3.4. Fields of Application

Table 1 summarizes the information on the studies included in this review.

### 3.5. Osteoarthritis

Four articles [31,32,33,34] investigated the potential of EVs derived from PRP as a therapeutic approach for osteoarthritis (OA). The efficacy was assessed in vitro in all cases. Additionally, the in vivo effect was also examined by two studies [31,34]. Although the experimental systems were different, the isolation methods and the EVs concentration were not the same, and they were obtained from both activated and non-activated PRP, the conclusion reached by all these authors was the same: PRP-derived EVs present a novel therapy for OA.

One major driver of OA is inflammation; therefore, any tool that aims at tackling this disease must be directed at relieving this inflammatory context. In this sense, Liu et al. [31] reported in a model consisting of IL-1β-treated chondrocytes resembling OA, PRP-EVs (PRP-Exo) significantly decreased the apoptotic rate of OA chondrocytes and presented the ability to reverse the typical increase in β-catenin, RUNX2 and Wnt5a associated to IL-1β-treated chondrocytes. In agreement with these authors, Otahal et al. [32] reported PRP-Exo anti-inflammatories effects in acute OA patient-derived chondrocytes. Another study by the same group [33] used a co-culture model involving primary chondrocytes exposed to activated macrophages to mimic the inflammatory environment in an OA joint. A lower proinflammatory cytokine profile was associated with plasma-derived EVs treatment in comparison with the complete blood products. Zhang et al. [34] also showed that PRP-Exo inhibited inflammation-induced chondrocyte degeneration. Concerning the in vivo models, Liu et al. [31] described the Wnt/β-catenin signaling pathway as a potential mechanism of action of platelet-rich plasma-derived exosomes (PRP-Exos) for OA therapy. In addition to reporting a protective role against IL-1β-induced apoptosis and degeneration of chondrocytes by PRP-Exo in vivo, Zhang et al. [34] confirmed that incorporating PRP-Exo into a thermosensitive hydrogel increased the local retention of exosomes thus delaying the development of subtalar osteoarthritis (STOA).

Additionally, the effect of PRP-derived EVs was also compared to other types of blood products. In this regard, Liu et al. [31] did so with respect to an activated PRP (PRP-As), concluding that the effect of PRP on alleviating OA may take place through PRP-Exos. On the other hand, Otahal et al. [32,33], under the assumption that cell composition differs among different blood products, investigated the role of EVs isolated from PRP and from hyperacute serum (hypACT), a serum-based blood product free of platelets, leukocytes, and fibrin. They reported differences in EVs populations between plasma (PRP) and serum (hypACT) derivatives regarding concentration and origin. The total EV concentration was significantly higher in the citrate-anticoagulated platelet-rich plasma (CPRP) [32,33], but only 44% were of platelet origin [32]. The authors concluded that CPRP EVs might provide beneficial anti-inflammatory effects in acute OA, whereas hypACT EVs would be more suitable for promoting regeneration in chronic OA [32].

### 3.6. Tissue Regeneration

Four in vitro studies [35,36,37,38] have been gathered in this general group. Two of them performed a biochemical and morphological characterization of PRP-derived exosomes after activation by different agonists [35] or by calcium chloride (CaCl_2_) [36] that have been already addressed in the previous section: *“PRP obtaining process affects extracellular vesicles production”.* The remaining two studies evaluated two different techniques, high-sensitivity flow cytometry [37] and cryo-electron microscopy [38] for EV characterization. In order to enhance the sensitivity of the conventional bench top flow cytometers, Stoner et al. [37] developed a custom device. They stimulated PRP with the ionophore A21387 to induce platelet EV release. The use of this calcium ionophore led to an increase in the fraction of annexin V and CD61-positive EVs. They reported that this high-sensitivity flow cytometry was able to detect individual EVs in plasma and EV sub-populations expressing cell surface markers, thus representing an alternative to light scatter-based EV detection devices. Yuana et al. [38] stated that morphological information on EV in fresh plasma is limited due to the limitations of the current analytical methods. Therefore, they studied the cryo-EM technique to support the EV analysis by other technologies. The authors concluded that cryo-EM allowed the visualization and characterization of individual EV in their native state in fresh human plasma and that most of the particles identified were lipoproteins thus recommending fasting to limit this non-EV lipidic structures contamination in the collection of blood samples.

### 3.7. Diabetic Retinopathy

Two studies from the same group fall into this category. Firstly, Zhang et al. [39] investigated the effects of PRP-Exos on retinal endothelial injury in diabetic rats and human retinal endothelial cells (HRECs) in vitro. Exosomes were obtained from activated PRP. The authors did not specify the type of activator they used for the normal PRP-Exos, whereas high glucose (HG) was employed for platelet activation in the in vitro culture system used for HG-PRP-Exos obtention. The authors reported that high glucose increased PRP-Exo levels in early diabetic retinopathy (DR). According to the results, the activation of the TLR4 pathway was involved in the role that PRP-Exos plays in this disease. The authors demonstrated that blocking this signaling pathway with TAK-242 could alleviate the PRP-Exo-induced negative effects, such as the decreased superoxide dismutase (SOD) activity, the increase production of malonyldialdehyde (MDA) and reactive oxygen species (ROS) and the blood-retinal barrier (BRB) dysfunction. They also suggested that the blockade of CXCL10 led to the TLR4 signaling pathway downregulation, thereby reducing PRP-Exo-induced adhesion molecules expression and preserving PRB function, thus it can be used as a therapeutic target for DR. Regarding this disease, Zhang et al. [40] conducted another in vitro study. They studied the effects of PRP-Exos on the fibrogenic activity of human retinal Müller cells (hMCs) by isolating PRP-Exos from the plasma of diabetic rats (DM-PRP-Exos) and normal control rats (Nor-PRP-Exos). As in the previous study, the authors also found an increase concentration of exosomes in the DM-PRP-Exos compared with that of the Nor-PRP-Exos. The analysis of the exosome cargo revealed enrichment of PDGF, bFGF, and TGF-β in the DM-PRP-Exos group. Additionally, they reported that DM-PRP-Exos significantly increased the proliferative and migratory ability of hMCs and the expression of CTGF and fibronectin. The authors also investigated the underlaying molecular mechanism, and they concluded that under hyperglycaemic conditions, the profibrogenic activity of PRP-Exos is mediated by YAP activation. They also demonstrated that DM-PRP-Exos activated this signaling pathway and enhanced both proliferative and fibrogenic activity of hMCs via PI3K/Akt pathway.

### 3.8. Intervertebral Disc Degeneration

The effect of PRP-derived EVs in intervertebral disc degeneration (IVD) has also been examined in two of the articles. Both studies concluded that PRP-EVs were effective in ameliorating IVD, albeit through different mechanisms of action. H_2_O_2_ was used to mimic the IVD conditions in vitro. Firstly, Xu et al. [42] compared the effects of PRP-Exo and reactive oxygen species scavenger (NAC) to revert the detrimental consequences of H_2_O_2_ addition on nucleus pulposus (NP) cells. Both treatments suppressed ROS and pro-inflammatory cytokines generation in NP cells treated with H_2_O_2_. The authors also identified the Keap1-Nrf2 pathway as the underlying mechanism by which PRP-Exo exerted its protective effects. PRP-Exo was enriched in miR-141-3p that activates the Keap1-Nrf2 pathway by degrading Keap1 and leading to the release of Nrf2 from the Keap1-Nrf2 complex, which finally results in a translocation from cytoplasm to the nucleus to trigger its antioxidant role. Dai et al. [41] compared platelet-derived EVs (PEV) vs. platelets (PLTs) from PRP using the same previous in vitro model (H_2_O_2_ addition), but with the double EVs concentration (100 µg/mL) and in cells of non-human origin. They also observed that PEV reduced oxidative stress and the inflammatory response. In fact, activation of PEV did not modify inflammatory factor levels associated with these two responses. Additionally, the results confirmed that PEVs could inhibit apoptosis and senescence and restore the metabolism of NP cells by increasing anabolic protein levels (Col2α, Sox9, and ACAN) and reducing the catabolic (MMP3, MMP13, and ADAMTS5). Moreover, they demonstrated that the SIRT1-PGC1α-TFAM pathway was involved in the PEV’s ability to maintain mitochondrial function. These authors also evaluated the therapeutic effects of PEV in a rat IVD model. Overall, the results demonstrated that PEVs could retard the progression of IVD in vivo. The data indicated that PEVs could restore the water content of NP, suppress matrix degradation, attenuate ROS levels and promote mitochondrial function, whereas platelets did not exert such a significant effect.

### 3.9. Muscle Injury

Two articles were included in this category. Catitti et al. [43] performed an in vitro study to analyze the cargo of PRP-EVs from athletes recovering from injuries by label-free proteomics. Despite the authors reporting to remove the leukocyte fraction from PRP, they claimed that leukocyte-derived EVs were the most represented EV subpopulation, followed by platelet derived-EVs and EVs from endothelium. The shotgun proteomic analysis revealed that the cargo of PRP-EVs was related to the regeneration process. One hundred and five proteins were identified, of which 32% were associated with “defense and immunity” biological function. These proteins were also involved in vesicle-mediated transport and wound-healing events. The authors suggested that platelet-derived EVs might be behind the regenerative potential of PRP. In turn, Iyer et al. [44] evaluated the effect of activated PRP-derived exosomes on the recovery of muscle function after injury. An in vivo rat model of muscle strain injury was used. They also compared the effect with MSC-derived exosomes. According to the published data, similar recovery of contractile function was reported for both groups compared to the control. Centrally nucleated fibers (CNFs) (a muscle regeneration marker) were also significantly increased after treatment with exosomes from both origins. However, genes involved in skeletal muscle regeneration were differentially affected. The *Myogenin* gene was significantly upregulated in muscles treated with PRP-exos, whereas MSC-exos treatment significantly reduced *TGF-β* expression. The authors concluded that exosomes derived from both origins could enable recovery after a muscle strain injury.

### 3.10. Wound Healing

Regarding cutaneous wound healing, two articles were included. Both in vitro and in vivo experiments were conducted. Despite the differences between these two studies (such as the experimental system or origin and composition of PRP), both reached the same conclusion that PRP-Exos could promote wound healing. Guo et al. [45] utilized the supernatant of activated PRP (PRP-AS) in order to compare the effect of PRP-Exos with PRP. Human microvascular endothelial cell line (HMEC-1) and primary dermal fibroblasts were established as the experimental system for the in vitro assays, while a diabetic rat model was used for the in vivo. The authors reported that PRP-Exos stimulated the proliferation and migration of HMEC-1 cells and fibroblasts to a greater extent than PRP. PRP-Exos also promoted a more significant tube formation in vitro. Similar effects were found in vivo. Moreover, the results showed that activation of Erk and Akt signaling pathways might underlie the PRP-Exos-induced angiogenesis. They also suggested that the effect of PRP-Exos on reepithelization could be mediated by increasing collagen synthesis through YAP activation. All these data stated that a substantial part of the PRP effects were mediated by exosomes. Despite Xu et al. [46] using immortalized keratinocytes (HaCaT cells) to evaluate the effects of PRP-Exos in vitro, they found a similar effect; that is, they also reported that these EVs effectively enhanced the proliferation, migration, and wound healing of these cells. This treatment was also superior to PRP treatment. However, the authors suggested a different underlying mechanism involved in this PRP-Exos-mediated wound healing. The authors demonstrated that USP15, a key mediator of protein deubiquitination, was detected at significantly higher levels in PRP-Exos and enhanced HaCaT cell functionality by promoting EIF4A1 deubiquitination. Moreover, according to the published data, this mechanism was also behind the in vivo promotion of cutaneous wound healing by PRP-Exos, as demonstrated by USP15 knockdown.

### 3.11. Miscellanea

#### 3.11.1. Cartilage Regeneration

Another in vitro study evaluated the effects of PRP-exos in the chondrogenic differentiation of bone marrow mesenchymal stem cells (BMSCs) from rats. They also assessed the combination with berberine (Exos-Ber), an isoquinoline alkaloid with anti-inflammatory properties. Three different concentrations were studied (5, 25, and 50 µg/mL). Their findings indicated that both PRP-Exos and Exos-Ber significantly promoted the proliferation, migration, and chondrogenic differentiation of BMSCs. They also found an increase in the chondrogenesis-related proteins Collagen II, SOX9, and ACAN. The greatest effect on all these biological processes was observed with the combination of exosomes and berberine. The results also suggested that Exos-Ber treatment exerted its role via the Wnt/β-Catenin pathway.

#### 3.11.2. Dental Pulp Regeneration

Bagio et al. [48] performed an in vitro analysis to determine the effect of different concentrations (0.5, 1, and 5%) of the thrombin-activated platelet-derived exosomes (T-aPDE) obtained from a PRP in the dental pulp regeneration. According to the authors, the 5% T-aDPE group promoted a greater human dental pulp stem cells (hDPSCs) viability rate and migration activity than the rest of the studied groups. The results also showed that mitochondria function improved after cells were cultured with 5% T-aDPE compared to the control groups. Finally, and regarding the angiogenesis process, this study demonstrated that hDPSCs treated with 5% T-aDPE showed a significantly increased in VEGF-A compared to the rest of the control groups and treatments after 24 h and 72 h.

#### 3.11.3. Hair Loss

Nilforoushzadeh et al. [49] compared the in vitro effects of different concentrations of exosomes derived from two different origins: adipose stem cells (hASCs) and platelet-rich plasma (PRP) for hair loss treatment. Preserving hair follicle inductivity of dermal papilla cells (DPCs) is the main concern in hair follicle regeneration. The authors reported that ASC-Exo significantly increased migration, proliferation, and amount of alkaline phosphatase (ALP), Versican, and alpha-smooth muscle actin (α-SMA) compared to other experimental groups. The 100 ug/mL concentration had the best biological effect.

#### 3.11.4. Osteonecrosis

Tao et al. [50] focused on the effect of PRP-Exos on the osteonecrosis of the femoral head (ONFH), a significant side effect of glucocorticoid (GC) use. Dexamethasone (DEX)-treated in vitro cell model and methylprednisolone (MPS)-treated in vivo rat model were used. The results showed that PRP-Exos treatment reverted the GC-induced apoptosis and antiproliferative effect both in vivo and in vitro. The authors concluded that the activation of the Akt/Bad/Bcl-2 signal pathway was responsible for this antiapoptotic role. Moreover, PRP-Exos could also rescue angiogenesis during DEX treatment and the co-administration with MPS-protected blood vessels in femoral heads in vivo. Similar results were found regarding osteogenesis, as the authors claimed that PRP-Exos could prevent the inhibition of osteogenesis mediated by DEX and promote the overregulation of osteogenesis-related proteins in vitro. This was further confirmed in vivo, where the cotreatment with PRP-Exos drastically reduced the GC-induced ONFH in rats. The authors concluded that PRP-Exos maintained the osteogenic differentiation and osteogenesis through the Wnt/β-catenin signaling pathway.

## 4. Discussion and Future Perspectives

According to all reported works compiled in this review, PRP-EVs represent a promising therapeutic approach in the field of tissue repair and regeneration. PRP-EVs promoted cell proliferation and migration [31,34,35,40,41,42,45,46,47,48,49,50] and angiogenesis [35,45,50] while reducing the inflammatory response [31,32,33,41,42], apoptosis [31,34,41,42,50] the oxidative stress [41] and senescence [41], among others. PRP-EVs seem to exert their function through different signaling pathways such as Wnt/β-catenin [31,47,50], AKT/ERK [35,45], TLR4 [39], PI3K/Akt [40], PGC1α-TFAM [41], Keap1-Nrf2 [42], YAP [45] and Akt/Bad/Bcl-2 [50]. All these effects may be due to their unique features, such as their lower immunogenicity, their ability to protect from degradation, their involvement in intercellular communication thanks to their composition, their ability to cross biological barriers, and the lack of need for expansion associated with other cellular sources [14,21]. However, from a scientific point of view, the lack of standardization in both the field of PRP and the EVs obtaining process makes it extremely challenging to compare results among studies. In this sense, the great variability in the PRP composition and obtaining procedures, the need for platelet activation or not and, if applicable, the type of agonist, the different EVs’ isolation and storage protocols, the absence of specific and universal markers, and the lack of consensus on the dose, among others, may have a direct impact on the yield, purity and cargo composition [21,64] (Figure 6). In addition, a large amount of information is omitted or not specified in the articles, which not only makes it impossible to compare but also to reproduce. Moreover, the International Society for Extracellular Vesicles, in its 2018 updated guideline [18], suggests a number of mandatory points concerning nomenclature, collection, characterization, functional studies, and reporting, which have not been followed in all the articles, notably the quantitative comparison of functional activity of EVs vs. EV-depleted fluid which none of the studies have conducted. Therefore, the resolution of all these issues is an essential requirement for performing translational studies.

The effect of PRP-EVs compared to that of PRP was conducted in 9 out of 20 articles included in this review [31,32,33,35,36,38,45,46,48]. In general, better performance was obtained with the former, which might suggest it could be a plausible alternative to replace the application of PRP for regenerative purposes. However, as previously mentioned, functionality studies depleting EVs should be carried out to confirm the contribution of these EVs in the PRP role. Furthermore, we should keep in mind that PRP is not only a source of proteins and growth factors, but it can also provide an autologous 3D fibrin matrix, which is necessary, in many cases, to support tissue regeneration. In this context, this naturally occurring scaffold provides physical support and mediates cell behavior [73]. Therefore, its critical biological role in tissue engineering cannot be ignored.

Their stability, extensive cargo-cell type and state-dependent, and their unique ability to cross the blood-brain barrier confer EVs as a promising powerful non-invasive tool for drug delivery and use as disease biomarkers [74,75]. Several review articles have indeed deeply addressed their role in several disease diagnoses, including autoimmune diseases [76,77], cardiovascular diseases [78,79,80], and neurodegenerative diseases [81,82,83]. Moreover, two clinical trials have been registered to evaluate the potential of PRP-EVs, both in chronic otitis media, one of which is still recruiting (NCT04761562), while the other is completed (NCT04281901) and with results already published [84]. In the latter, the authors concluded that autologous platelet—and extracellular vesicles—rich plasma represents a novel and successful treatment for a chronically radical mastoid cavity when the standard methods have been exhausted. In this case, the authors further specified that the inclusion of leukocytes was avoided in the plasma preparation. Additional controlled clinical studies should also be performed to evaluate the real clinical relevance of this emerging approach.

In summary, the use of PRP-EVs is an emerging field of great interest that represents a promising therapeutic option, as their efficacy has been confirmed in the majority of fields of applications included in this review. Nevertheless, there is an urgent need to establish standardized conditions to ensure optimized and detailed protocols to define, among others, the obtaining process, the dose, or the origin. Further studies should also be performed to elucidate the real contribution of EVs to PRP. Additionally, the evaluation of an integrated strategy that takes advantage of the synergistic effects of combining PRP-derived fibrin and EVs can entail a leap forward in the development of new applications. Nevertheless, these promising results provide a novel basis to deal with the regenerative medicine and drug delivery fields in the future.

## Figures and Tables

**Figure 1 ijms-24-13043-f001:**
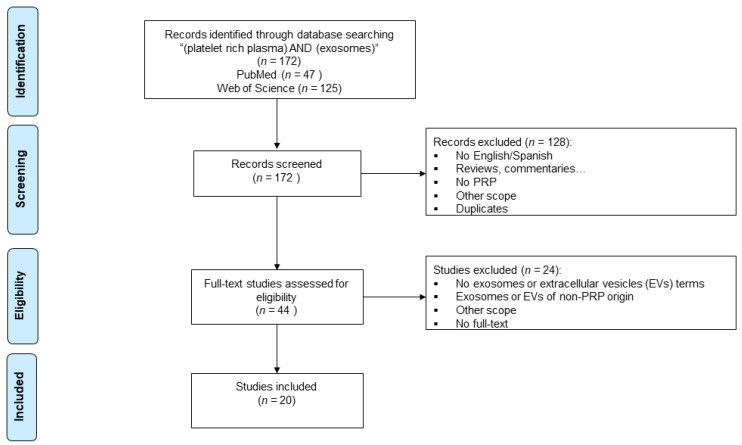
Flowchart summarizing the selection process following the Preferred Reporting Items for Systematic Reviews and Meta-Analyses (PRISMA) guidelines.

**Figure 2 ijms-24-13043-f002:**
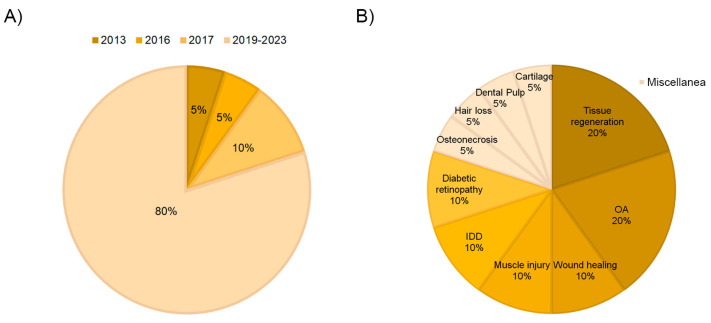
Distribution of the selected articles according to year of publication (**A**) and field of application (**B**).

**Figure 3 ijms-24-13043-f003:**
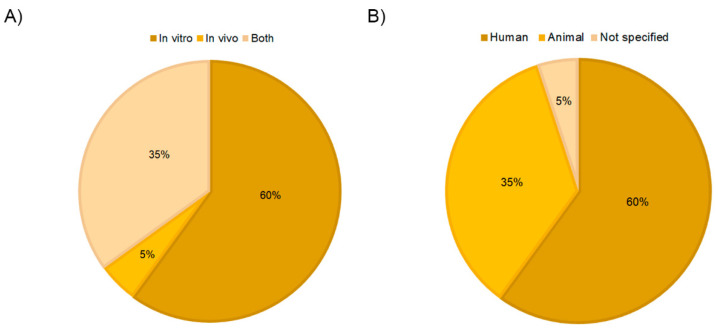
Distribution of the selected publications according to study type (**A**) and PRP origin (**B**).

**Figure 4 ijms-24-13043-f004:**
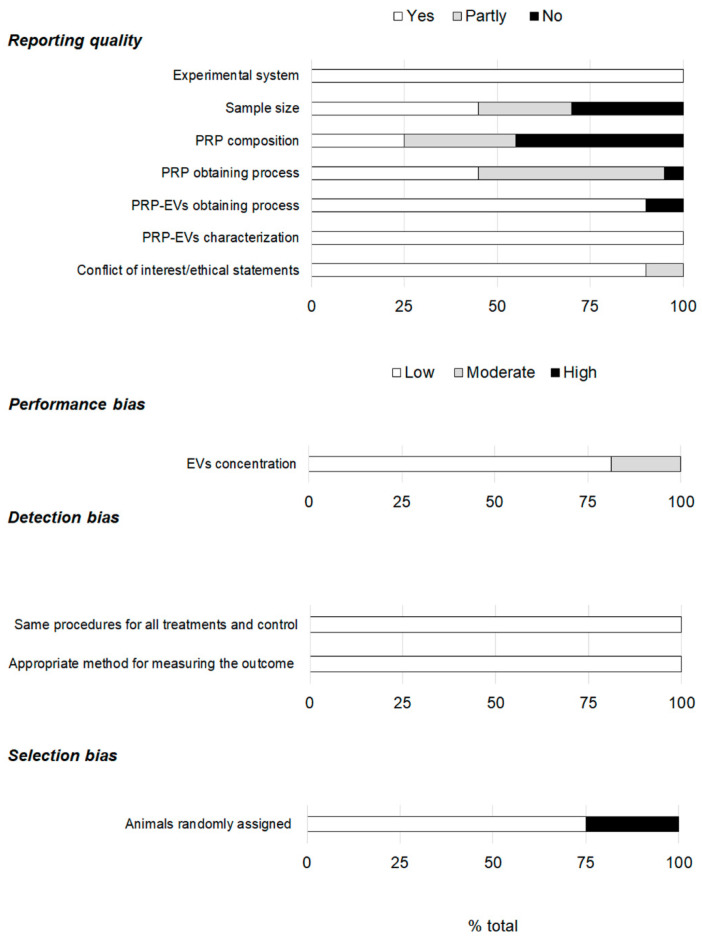
Assessment of the reporting quality and risk of bias.

**Figure 5 ijms-24-13043-f005:**
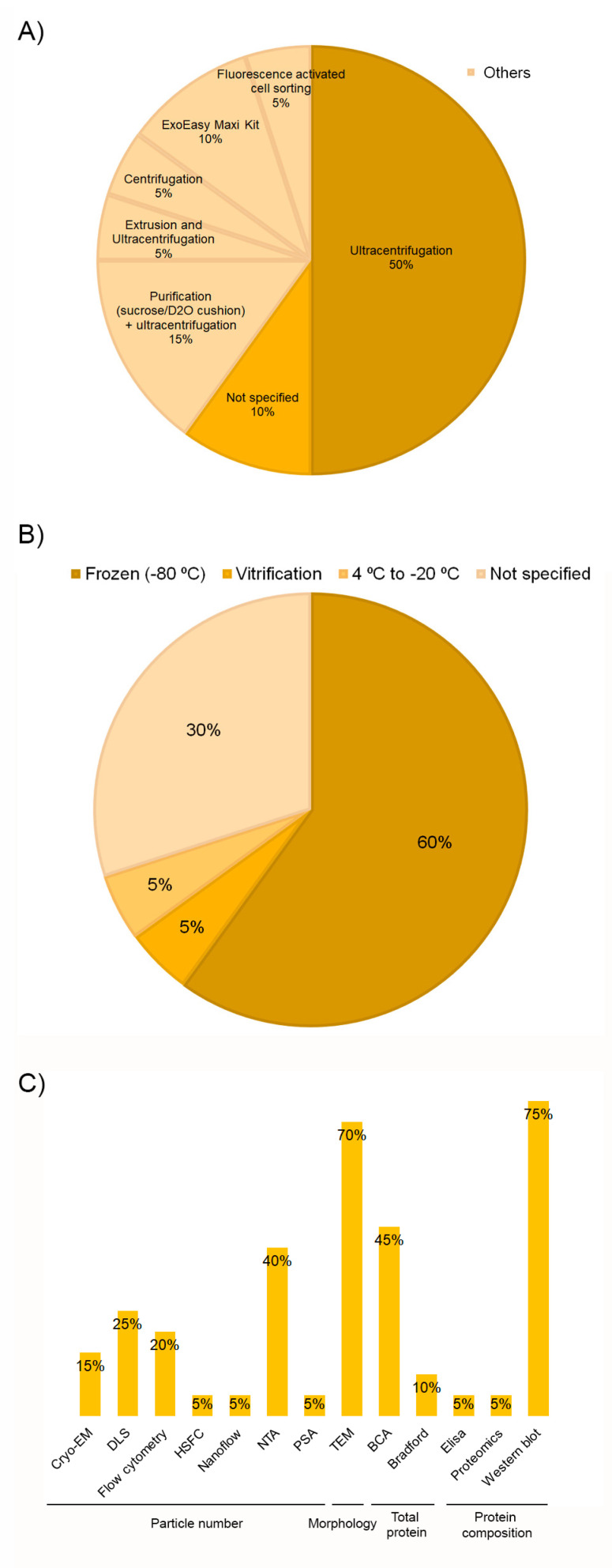
Distribution of the selected publications according to EVs isolation methods (**A**), storage temperature (**B**), and characterization methods (**C**).

**Figure 6 ijms-24-13043-f006:**
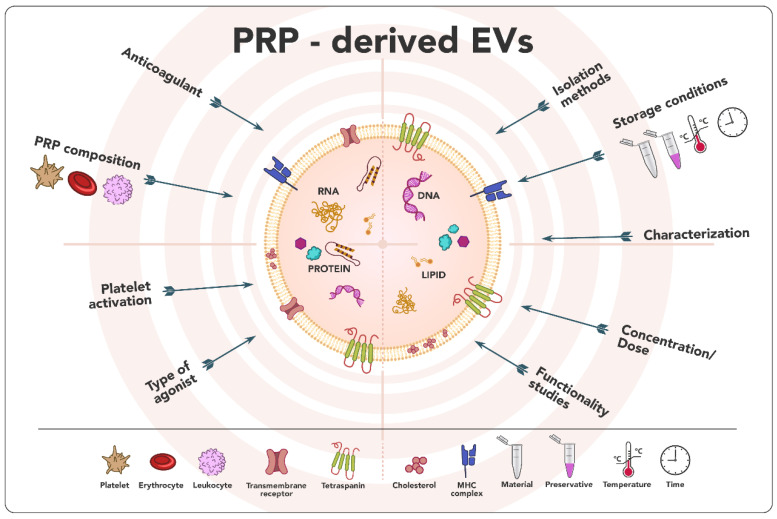
Issues needed to be elucidated in order to promote the clinical translation of PRP-derived EVs.

**Table 1 ijms-24-13043-t001:** Summary information of the studies included in this review.

Ref.	Field of Application	Study Type	Experimental System	Comparison Groups	[Exosomes]	Methods	Summary Results
Liu et al., 2019 [31]	Osteoarthritis	In vitro	Primary rabbit chondrocytes	**Group I**: PBS **Group II**: IL-1β **Group III**: IL-1β + PRP-exos **Group IV**: IL-1β + PRP-As	5, 50 µg/mL	Elisa, proliferation, apoptosis, migration, scratch wound assay, western blotting	PRP-Exos and PRP-As both inhibited TNF-α release. PRP-Exos could significantly decrease the apoptotic rate of OA chondrocytes compared with PRP-As. PRP-Exos promoted proliferation and migration to a larger extent than PRP-As. The potential mechanism might be through activation of the Wnt/β-catenin signaling pathway. β-catenin, RUNX2, and Wnt5a typical increase in IL-1β-treated chondrocytes could accordingly reverse by both PRP-Exos and PRP-As, although the former performed better than the latter. In vivo, both PRP-Exos and PRP-As prevented the progression of OA, the effect of PRP-Exos being significatively more effective.
		In vivo	Rabbit knee OA model	**Group I**: Control **Group II**: OA **Group III**: OA + PRP-Exos **Group IV**: OA + PRP-As	100 µg/mL	Histological analysis, immunohistochemistry, the OARSI score system	
Otahal et al., 2020 [32]	Osteoarthritis	In vitro	Human chondrocytes from OA cartilage	**Group I**: CPRP **Group II**: HypACT **Group III**: CPRP-EVs **Group IV**: HypACT-EVs **Group V**: 10% FCS (Control)	hypACT EV: 1.42 × 10^9^ ± 2.12 × 10^6^ EVs per well CPRP EV:1.42 × 10^9^ ± 2.95 × 10^7^ EVs per well	Western blot, RT-qPCR, ELISA	The total EV concentration was significantly lower in hypACT than in CPRP. Around 84% of total EVs originated from platelets in hypACT, whereas platelet EVs constituted a 44% proportion of total EVs in CPRP. The clotting for hypACT obtention might be responsible for such differences between plasma (PRP) and serum (hypACT) derivatives. EV treatment of OA patient-derived chondrocytes enhanced expression of both anabolic (*Type II collagen*, *SOX9*, and *ACAN*) and catabolic (*MMP3*) chondroprotective markers, compared to full blood products. HypACT EVs prevented *Type I collagen* expression. CPRP EVs might provide beneficial anti-inflammatory effects in acute OA, whereas hypACT EVs might be more suitable to drive chondrogenesis and cartilage regeneration in chronic OA.
Otahal et al., 2021 [33]	Osteoarthritis	In vitro	Human primary OA chondrocytes. Human primary monocytes and THP-1 were differentiated and activated into M1 macrophages. Coculture.	**Group I**: CPRP **Group II**: HypACT **Group III**: CPRP-EVs **Group IV**: HypACT-EVs **Group V**: 10% FCS (Control)	hypACT EV: 1.91 × 10^8^ ± 1.09 × 10^8^ EVs CPRP EV: 1.91 × 10^8^ ± 1.65 × 10^8^ EVs of resuspended EV pellets added to 2 mL co-culture medium	RT-qPCR, ELISA	Mode sizes were similar between EVs from CPRP and hypACT. Higher concentrations of EVs were obtained from CPRP compared to hypACT. EV treatment was associated with a reduction in proinflammatory cytokine levels compared to blood product treatment. EV treatment increased *Type II collagen* expression while lowering TNF-α and IL-1β secretion. This could indicate either that the EV treatment replicates the effect of the complete blood product or that the EV concentration from blood products was too low to elicit therapeutically relevant changes. This did not exclude the possibility that EVs are therapeutically relevant agents in blood product therapy.
Zhang et al., 2022 [34]	Subtalar Osteoarthritis	In vitro	mBMSCs and murine chondrocytes	**Group I**: Control **Group II**: Gel **Group III**: Exo-gel	The conditioned medium from Exo-Gel (200 μg Exo incorporated in 100 μL thermossensitive Gel)	Uptake of Exo, viability, cell proliferation, migration assay, RTqPCR, chondrocyte apoptosis, cell fluorescence analysis, Western blot.	PRP-Exo incorporated into gel (Exo-gel) could release and promote the proliferation and migration of mBMSCs and chondrocytes, enhance the chondrogenic differentiation of mBMSCs, and inhibit inflammation-induced chondrocyte degeneration. PRP-Exo had positive regulation of mBMSCs function and played a protective role against IL-1β-induced apoptosis and degeneration of chondrocytes in vivo. Application of this system allowed retention of EVs (Exo) in the ankle compared to Exo-only injection and maintained the local concentration of Exo. Together, Exo-Gel had effective effects on suppressing cartilage degeneration and the development of STOA.
		In vivo	STOA model	**Group I**: PBS **Group II**: Gel **Group III**: Exo **Group IV**: Exo-Gel	4 µg/2 µL (2000 µg/mL)	Histomorpholgical detection, immunohistochemical analysis, TUNEL staining	
Rui et al., 2021 [35]	Tissue regeneration	In vitro	HUVECs	**Group I**: PRP-AS **Group II**: PRP-Exos + saline **Group III**: PRP-Exos + thrombin **Group IV**: PRP-Exos + Ca^2+^ **Group V**: PRP-Exos + mixture	50 µg/mL	Exosomes internalization, cell proliferation, migration, matrigel tube formation, Western blot	The size of calcium-activated PRP-Exos was larger than the thrombin-activated group and the mixture-activated group. PRP activated by the mixture released the highest concentration of exosomes. All the activation methods were suitable for PRP-Exos isolation, calcium gluconate alone was found to be weaker. The protein levels of VEGF, PDGFBB, bFGF, TGF-β in the mixture-activated PRP-Exos group were higher than in other groups. PRP-Exos yielded after thrombin and calcium gluconate together were found to contain more cytokines than the other groups. PRP-Exos harvested after activation by thrombin and calcium gluconate together could more significantly promote HUVECs proliferation, migration, and formation of vessel-like via the AKT ERK signal pathway, compared with other groups.
Saumell-Esnaola et al., 2022 [36]	Tissue regeneration	In vitro	N/A	**Group I**: PLT **Group II**: PLT-Ca^2+^ **Group III**: PLT-Exos **Group IV**: PLT-Exos-Ca^2+^ **Group V**: PPP **Group VI**: PRP	N/A	Semiquantitative measurement of 105 analyte levels, including cytokines, chemokines, growth factors, and other soluble proteins	The calcium activation of PRP promoted the release of highly purified platelet-derived exosomes, showing a concordant size and morphology and the absence of contaminants from other cellular compartments. Calcium was proved to alter the cytokine cargo expression profile of PLT-Exos-Ca^2+^ in relation to exosomes isolated from non-activated platelets. Although PRP calcium activation promoted exosome release, its net contribution to the total PRP might be minimal, in view of the low yield of exosome protein content relative to the total in PRP.
Stoner et al., 2016 [37]	Tissue regeneration	In vitro	N/A	N/A	N/A	HSFC/VFC	Treatment of platelet-rich plasma with calcium ionophore resulted in an increase in the fraction of annexin V and CD61-positive EVs. Vesicle flow cytometry using fluorescence-based detection of EVs had the potential to realize the potential of cell-derived membrane vesicles as functional biomarkers for a variety of applications.
Yuana et al., 2013 [38]	Tissue regeneration	In vitro	N/A	**Group I**: PRP **Group II**: PPP **Group III**: PFP **Group IV**: PRP-EVs **Group V**: PPP-EVs **Group VI**: PFP-EVs	N/A	cryo-EM/cryo-ET	EVs isolated from fresh plasma were highly heterogeneous in morphology and size. EV constituted only a very small fraction of all particles present in fresh plasma isolated from the blood of fasted healthy volunteers, whereas the majority were lipoprotein particles. Cryo-EM was a powerful technique that enabled the characterization of EV in fresh plasma.
Zhang et al., 2019 [39]	Diabetic retinopathy	In vitro	HRECs	**Group I**: ctrl **Group II**: ctrl-PRP-Exos **Group III**: DM-PRP-Exos TLR4 assay: **Group I**: without PRP-Exos (Ctrl) **Group II**: with normal PRP-Exos **Group III**: HG-PRP-Exos **Group IV**: HG-PRP-Exos + TAK-242 CXCL10 assay: **Group I**: without PRP-Exos (Ctrl) **Group II**: HG-PRP-Exos **Group III**: HG-PRP-Exos + anti-CXCL10	500 μg PRP-Exos (RNA concentration) suspended in 500 μL PBS (1000 µg/mL)	Immunohistochemical measurements, western blot, ROS	The level of circulating PRP-Exos was greatly elevated in diabetic rats. PRP-Exos mediated hyperglycemia-induced retina endothelia injury via TLR4 signaling pathway. PRP-Exos can activate the TLR4 pathway by promoting an increase in TLR4 and its downstream proteins in vitro and in vivo. CXCL10 was a major effector of PRP-Exo-derived retinal endothelial damage; thus, blocking PRP-Exo-derived CXCL10 may be a novel therapeutic approach for DR.
		In vivo	Diabetes mellitus induced in rats	**Group I**: normal controls were treated with vehicle (PBS) **Group II**: DM (diabetic rats were treated with vehicle) **Group III**: DM + TAK.	NS	Immunohistochemical measurements, western blot, ROS, measurement of blood-retinal barrier (BRB) breakdown	
Zhang et al., 2020 [40]	Diabetic retinopathy/Retinal fibrogenesis	In vitro	hMCs	**Group I**: Nor-PRP-Exos **Group II**: DM-PRP-Exos **Group III**: Nor-PL (control) **Group IV**: DM-PL (control)	500 μg PRP-Exos (RNA concentration) suspended in 500 μL PBS (1000 µg/mL)	Internalization, cell proliferation, migration, Elisa, Western blot, Immunohistochemical analysis	The concentration of exosomes was increased in the DM-PRP-Exos group compared with the Nor-PRP-Exos group. Analysis of the exosome cargo showed that the levels of PDGF, bFGF, and TGF-β were elevated in DM-PRP-Exos compared with those in PL or Nor-PRP-Exos. DM-PRP-Exos significantly increased the proliferative and migratory ability of hMCs and the expression of CTGF and fibronectin. DM-PRP-Exos contributed to activating YAP and promoting the fibrogenic activity of Müller cells via the PI3K/Akt pathway.
Dai et al., 2023 [41]	Intervertebral disc degeneration	In vitro	NP cells from rats	**Group I**: Normal control **Group II**: H_2_O_2_ **Group III**: PLTs (H_2_O_2_ + 100 μg/mL platelets) **Group IV**: PEVs (H_2_O_2_ + 100 μg/mL PEVs)	100 μg/mL	Cytotoxicity and proliferation, cytokine detection, ROS detection, apoptosis, SA-β-gal staining, PEVs uptake and mitochondrial localization, qPCR, western blot, immunofluorescence staining, mitochondrial function, ATP detection, lactic acid detection	A significant increase in IL-1β, IL-6, and TNFα production in platelets after thrombin addition was detected. PEVs did not release these pro-inflammatory factors. PEVs inhibited apoptosis and senescence in NP cells. PEVs could restore impaired mitochondrial function, reduce oxidative stress, and restore cell metabolism by regulating the SIRT1-peroxisome PGC1α-TFAM pathway. In rat models, PEVs retarded the progression of IVDD by reducing oxidative stress and inflammatory response.
		In vivo	A rat IVDD model	**Group I**: NC group **Group II**: Puncture + vehicle (PBS) **Group III**: Puncture + PLTs **Group IV**: Puncture + PEVs	100 μg/mL	Micro-CT and MRI, histological analysis, immunofluorescence staining, systemic toxicity	
Xu et al., 2021 [42]	Intervertebral disc degeneration	In vitro	NP cells and HEK293T cells	**Group I**: Control **Group II**: H_2_O_2_ **Group III**: H_2_O_2_ + PRP-exo **Group IV**: H_2_O_2_ + NAC	50 µg/mL	RTqPCR, Western blot, Elisa, ROS, cell apoptosis, cell proliferation and viability, Dual-luciferase reporter gene system assay, RNA pull-down assay	Both PRP-exo and NAC suppressed ROS generation and pro-inflammatory cytokines in NP cells treated with H_2_O_2_. PRP-exo and NAC ameliorated H_2_O_2_- induced cell apoptosis in NP cells. PRP-exo reversed the detrimental effects of H_2_O_2_ treatment on NP cells. The Keap1-Nrf2 pathway could be activated by PRP-exo in the NP cells. PRP-exo delivered miR-141-3p to degrade Keap1, leading to the release of Nrf2 from the Keap1-Nrf2 complex, which further translocated from cytoplasm to nucleus to exert its anti-oxidant effects, resulting in the attenuation of IVD degeneration.
Catitti et al., 2022 [43]	Muscle injury	In vitro	N/A	PRP from athletes recovering from injuries	N/A	EV Protein Cargo Detection by Label-Free Proteomics	Leukocyte-derived EVs were the most represented EV subpopulation within PRP-EVs, followed by platelet-derived EVs and EVs from endothelium. There were 105 proteins identified, mostly classified in the “defense and immunity” biological function and related to “vesicle-mediated transport” and “wound healing”.
Iyer et al., 2020 [44]	Muscle injury	In vivo	Rat bone marrow–derived MSCs/Animal model of muscle strain injury	**Group I**: Saline **Group II**: PRP exosomes **Group III**: MSC exosomes	1 × 10^8^ exosomes resuspended in 50 µL of PBS	Recovery of muscle function, histological assay to detect myogenesis, RT-PCR	Both PRP-exos and MSC-exos accelerated recovery of contractile function over the saline-treated group. A significant increase in CNFs was observed with both types of exosome groups by day 15. Muscles treated with PRP-exos had increased expression of Myogenin gene. Exosomes derived from PRP or MSCs could facilitate recovery after a muscle strain injury in a small-animal model.
Guo et al., 2017 [45]	Wound healing	In vitro	HMEC-1 and primary dermal fibroblasts treated with Y-27632 2HCl, or transfected with shYAP or S127A	**Group I**: PRP-AS **Group II**: PRP-Exos.	5, 50 µg/mL	Cell proliferation, PRP-Exos-loaded sodium alginate hidrogel, PRP-Exos release, cell migration, capillary-like construction activity, YAP nuclear localization, gene expression analysis, and western blotting	PRP-Exos increased the proliferation and migration of HMEC-1 cells and fibroblasts and tube formation to a greater extent than PRP. The wounds treated with PRP-Exos closed significantly faster. Higher blood vessel area, blood vessel number, and longer neo-epithelium in the PRP-Exos-treated defects were observed. Exosomes released by PRP may contribute to PRP-induced angiogenesis through activation of Erk and Akt signaling pathways, and PRP-induced re-epithelialization may be triggered by activation of YAP.
		In vivo	Diabetic rat model	**Group I**: Control **Group II**: SAH **Group III**: PRP **Group IV**: PRP-Exos	It was calculated on the basis of the determined PRP-AS protein concentration and mixed with SAH	Wound size, microCT, histological analysis, IHC and IF methods	
Xu et al., 2021 [46]	Wound healing	In vitro	HaCaT	**Group I**: Control (PBS) **Group II**: PRP **Group III**: PRP-Exos.	N.S.	Internalization of exosomes, cell proliferation, wound healing, migration, cell cycle analysis, Western blot, qPCR	PRP-Exo treatment enhanced in vitro keratinocyte responses. PRP-Exo treatment was superior to PRP treatment as a means of enhancing wound healing associated with increased USP15 in wound tissues relative to PBS or PRP administration.
		In vivo	Murine Cutaneous Wound Model	**Group I**: PBS **Group II**: PRP-Exos **Group III**: siRNA-NC **Group IV**: siRNA-USP15 **Group V**: siRNAUSP15+ PRP-Exos	PRP-exos: 100 μg PRP-Exos in 100 μL PBS (1000 µg/mL); siRNA-USP15 + PRP-Exos: 10μmol/L	Wound healing, histological analysis, and Immunohistochemical Staining	
Dong et al., 2022 [47]	Cartilage regeneration	In vitro	BMSCs from rats	**Group I**: PRP-exos **Group II**: Ber **Group III**: Exos-Ber **Group IV**: Control (PBS or medium)	5, 25 and 50 µg/mL	Proliferation, migration, chondrogenesis	PRP-exos or Exos-Ber at 25 μg/mL led to significantly enhanced proliferation of BMSCs. PRP-exos, Ber, and Exos-Ber could significantly promote the chondrogenic differentiation of BMSCs. The highest upregulation was observed with Exos-Ber. PRP-exos and Exos-Ber could significantly promote the migration of BMSCs. The proteins Collagen II, SOX9, and ACAN were upregulated in BMSCs exposed to Ber, PRP-exos, or Exos-Ber. The greatest upregulation was observed in cells treated with Exos-Ber. Exos-Ber could promote BMSCs chondrogenic differentiation via the Wnt/β-catenin signaling pathway.
Bagio et al., 2023 [48]	Dental Pulp Regeneration	In vitro	hDPSCs	**Group I**: no FBS **Group II**: 10% FBS **Group III**: 10% PRP-T **Group IV**: 0.5% T-aPDE **Group V**: 1% T-aPDE **Group VI**: 5% T-aPDE.	0.5%/1%/5%	Viability assay/Migration/VEGF-A Expression	The 5% T-aPDE group could produce hDPSCs with a superior mean viability rate compared with the other conditioned media culture groups, and it had the best migration activity of all the groups. The VEGF-A amount of hDPSCs escalated more significantly 24 to 72 h after they were cultured in 5% T-aPDE.
Nilforoushzadeh et al., 2021 [49]	Hair loss	In vitro	DPCs and ORSCs	**Group I**: Control (serum-free culture medium) **Group II**: hASCs-Exo **Group III**: PRP-Exo	25, 50 and 100 µg/mL	Internalization of exosomes, cell scratch, proliferation, SEM, RT-PCR	Exosomes could enter the DPCs cytoplasm and be localized in the perinuclear area. The culture of human DPCs with ASC-Exo exhibited significantly increased migration, proliferation, and hair inductivity compared to other experimental groups. When 100 μg/mL ASCs-Exo was compared to the same concentration of PRP-Exo, the former significantly promoted DP proliferation and migration, as well as overregulation of ALP, Versican, and α-SMA proteins.
Tao et al., 2017 [50]	ONFH	In vitro	MC3T3-E1 cells, HMEC-1 and primary BMSCs/DEX-treated in vitro cell model	**Group I**: Control **Group II**: DEX **Group III**: DEX + PRP-exos **Group IV**: PRP-Exos	50 µg/mL	Cell viability, cell proliferation, tube formation, migration, apoptosis, osteogenesis, Western blot.	PRP-Exos played proliferative and anti-apoptotic roles against GC-ER induced-stress in vitro and in vivo. The co-administration with PRP-Exos could rescue angiogenesis in vitro and had a notable protective effect on blood vessels in vivo. PRP-Exos could prevent the inhibition of osteogenesis observed in BMSCs and MC3T3-E1 cells caused by DEX in vitro and could rescue the expression of osteogenesis-related proteins. PRP-Exos showed to be effective in preventing GC-induced ONFH in rats. The stimulatory effects of PRP-Exos on anti-apoptosis mainly resulted from the activation of the Akt/Bad/Bcl-2 signal pathway. In bone cells, PRP-Exos rescued the osteogenic protein expression level through the Wnt/β-catenin signaling pathway.
		In vivo	MPS-treated in vivo rat model	**Group I**: Control group **Group II**: MPS group **Group III**: PRP-Exos group	100 μg exosomes (dissolved in 200 μL of PBS) (500 µg/mL)	Micro CT (angiography and osteogenesis), TUNEL assay and Ki67 immunostaining	

**ALP:** Alkaline phosphatase. **α-SMA:** Smooth muscle alpha-actin. **ATP:** Adenosine triphosphate. **Ber:** Berberine. **bFGF:** Basic fibroblast growth factor. **BRB:** Blood-retinal barrier. **CPRP:** Citrate-anticoagulated platelet-rich plasma. **Cryo-EM:** Cryo-electron microscopy. **Cryo-ET:** Cryo-electron tomography. **CTGF:** Connective tissue growth factor. **CXCL10:** C-X-C Motif Chemokine Ligand 10. **DEX:** Dexamethasone. **DM:** Diabetes mellitus. **DPCs:** Dermal papilla cells. **DR:** Diabetic retinopathy. **EVs:** Extracellular vesicles. **FBS:** Fetal bovine serum. **FCS:** Fetal calf serum. **GC-ER:** Glucocorticoids endoplasmic reticulum. **HaCaT:** Human immortalized keratinocytes. **hASCs-Exo:** Exosomes derived from human adipose stem cells. **hDPSCs:** Human dental pulp stem cells. **HEK293T:** Human Embryonic Kidney 293 that contains the SV40 large T antigen. **HG:** High glucose. **hMCs:** Human immortalized retinal Müller cells. **HMEC-1:** Human microvascular endothelial cell line. **HRECs:** Human retinal endothelial cells. **HSFC:** High sensitivity flow cytometer. **HUVECs:** Human umbilical vein endothelial cells. **HypACT:** Hyperacute serum. **IF:** Immunofluorescence. **IHC:** Immunohistochemical. **IL-1β:** Interleukin 1 beta. **IL6:** Interleukin 6. **IVDD:** Intervertebral disc degeneration. **Keap1-Nrf2:** Kelch-line ECH-associated protein 1-Nuclear factor (erythroid-derived-2)- like 2. **mBMSCs:** Mouse bone mesenchymal stem cells. **MC3T3-E1:** Murine osteoblastic cells. **Micro-CT:** Micro-computed tomography. **MPS:** Methylprednisolone. **MRI:** Magnetic resonance imaging. **MSCs:** Mesenchymal stem cells. **NAC:** N-Acetyl-D-cysteine (ROS scavenger). **Nor:** Normal. **NP:** Nucleous pulposus. **OA:** Osteoarthritis. **OARSI:** Osteoarthritis Research Society International. **ONFH:** Steroid-induced osteonecrosis of the femoral head (ONFH). **ORSCs:** Outer root sheath cells from the human scalp. **PBS:** Phosphate buffered saline. **PDGFBB:** Platelet-derived growth factor BB. **PEVs:** Platelet-derived extracellular vesicles. **PFP:** Platelet-free plasma. **PGC1α-TFAM:** Proliferator-activated receptor gamma coactivator 1α-mitochondrial transcription factor A. **PL:** Platelet lysate. **PLT:** Platelets. **PPP:** Platelet-Poor Plasma. **PRP:** Platelet-Rich Plasma. **PRP-As:** Activated platelet-rich plasma. **PRP-AS:** Supernatant of activated PRP. **PRP-exos:** Exosomes derived from platelet-rich plasma. **PRP-T:** Platelet-rich plasma-thrombin. **ROS:** Reactive oxygen species. **RT-PCR:** Reverse transcription-polymerase chain reaction. **RT-qPCR:** Reverse transcription-quantitative polymerase chain reaction. **RUNX2:** RUNX Family Transcription Factor 2. **SA-β-gal:** Senescence-associated β-gal. **SAH:** Sodium alginate hidrogel. **SEM:** Scanning electron microscope. **SIRT1:** Sirtuin1. **SOX9:** SRY-Box Transcription Factor 9. **STOA:** Subtalar osteoarthritis. **TAK-242:** Ethyl-(6*R*)-6-(N-(2-chloro-4-fluorophenyl)sulfamoyl)cyclohex-1-ene-1-carboxylate, (*R*)-Ethyl 6-(N-(2-chloro-4-fluorophenyl)sulfamoyl)cyclohex-1-enecarboxylate, (TLR4 inhibitor). **T-aPDE:** Thrombin-activated platelet-derived exosome (T-aPDE). **TFAM:** Mitocondrial transcription factor A. **TGF-β:** Transforming growth factor-β. **TNF-α:** tumor necrosis factor-α. **UC:** Ultracentrifugation. **USP15:** Ubiquitin Specific Peptidase 15. **VEGF:** Vascular endothelial growth factor. **VFC:** Vesicle flow cytometry. **Wnt5a:** Wnt Family Member 5A. **YAP:** Yes-associated protein. **Y-27632 2HCl:** Selective ROCK inhibitor.

**Table 2 ijms-24-13043-t002:** Description of the PRP obtaining process in the reviewed articles.

Article	PRP Origin	Anticoagulant	Leukocytes	Activator	EVs/Exosomes
Liu et al., 2019 [31]	Rabbit PRP	ACD-A	No	No	PRP-derived exosomes
Otahal et al., 2020 [32]	Human CPRP	Trisodium Citrate	Low numbers of monocytes (20 per µL)	No	CPRP-EVs
Otahal et al., 2021 [33]	Human CPRP	Trisodium Citrate	NS	No	CPRP-EVs
Zhang et al., 2022 [34]	Human PRP	ACD-A	NS	CaCl_2_ and thrombin	PRP-derived exosomes
Rui et al., 2021 [35]	Human PRP	Citrate glucose	No	Thrombin/Calcium gluconate/Mixture of both	PRP-derived exosomes
Saumell-Esnaola et al., 2022 [36]	Human PRP (13-00-11)	Sodium citrate	No	CaCl_2_	PRP Platelets derived exosomes
Stoner et al., 2016 [37]	Rat PRP	Sodium citrate	NS	Ionophore A23187	EVs
Yuana et al., 2013 [38]	Human PRP	Sodium citrate and EDTA	NS	No	PRP-EVs
Zhang et al., 2019 [39]	Rat PRP	Sodium citrate	NS	NS *,**	PRP-derived exosomes
Zhang et al., 2020 [40]	Rat PRP (from diabetic rats and from normal rats)	Sodium citrate	NS	NS *	PRP-derived exosomes
Dai et al., 2023 [41]	Human PRP	ACD-A	NS	w/o and w/Thrombin	Platelet-derived EVs
Xu et al., 2021 [42]	NS	NS	NS	NS	PRP-derived exosomes
Catitti et al., 2022 [43]	Human PRP	ACD-A	No	NS	PRP-EVs
Iyer et al., 2020 [44]	Rat PRP (Arthrex ACP Double Syringe System)	NS	No	Sonication	PRP-derived exosomes
Guo et al., 2017 [45]	Human PRP	ACD-A	No	NS *	PRP Platelets derived exosomes
Xu et al., 2021 [46]	Mice PRP	NS	Yes	NS	PRP-derived exosomes
Dong et al., 2022 [47]	Rat PRP	ACD-A	No	Sonication	PRP-derived exosomes
Bagio et al., 2023 [48]	Human PRP	NS	Yes	Thrombin	T-aPDE
Nilforoushzadeh et al., 2021 [49]	Human PRP (from cord blood)	NS	NS	NS	PRP-derived exosomes
Tao et al., 2017 [50]	Human PRP	ACD-A	No	NS *	PRP-derived exosomes

**ACD-A:** acid citrate dextrose solution A. **CPRP**: citrate-anticoagulated platelet-rich plasma. **EVs**: extracellular vesicles; **NS**: not specified. **PRP**: platelet-rich plasma. **T-aPDE**: thrombin-activated platelet-derived exosome. * The PRP was activated, but the type of activator was not specified. ** High glucose (HG) was also employed for the HG-PRP-Exos for some of the assays.

**Table 3 ijms-24-13043-t003:** EVs isolation methods and storage conditions.

Article	Isolation	EVs Storage			
		Type of Tube	Buffer	Temperature	Time
Liu et al., 2019 [31]	ExoEasy Maxi kit (Qiagen)	N.S.	PBS	Frozen (−80 °C)	N.S.
Otahal et al., 2020 [32]	Ultracentrifugation	N.S.	PBS	Frozen (−80 °C)	N.S.
Otahal et al., 2021 [33]	Ultracentrifugation	N.S.	PBS	Frozen (−80 °C)	N.S.
Zhang et al., 2022 [34]	Ultrafiltration/purification with 30% sucrose D2O cushion and ultracentrifugation	Ultra-clear^TM^ tube (Beckman Coulter)	PBS	Frozen (−80 °C)	N.S.
Rui et al., 2021 [35]	Gradient ultracentrifugation	N.S.	PBS containing protease inhibitor cocktail and phosphataseinhibitor cocktail	Frozen (−80 °C)	N.S.
Saumell-Esnaola et al., 2022 [36]	Differential ultracentrifugation	N.S.	PBS	Frozen (−80 °C)	N.S.
Stoner et al., 2016 [37]	N.S.	N.S.	N.S.	N.S.	N.S.
Yuana et al., 2013 [38]	Ultracentrifugation	N.S.	HEPES buffer	Vitrification	N.S.
Zhang et al., 2019 [39]	Ultracentrifugation	N.S.	PBS	Frozen (−80 °C)	N.S.
Zhang et al., 2020 [40]	Ultracentrifugation	N.S.	PBS	Frozen (−80 °C)	N.S.
Dai et al., 2023 [41]	Extrusion and ultracentrifugation	N.S.	PBS + EDTA + PGE1	N.S.	Test stability 0, 1, 3, 5, and 7 days (in vitro)
Xu et al., 2021 [42]	N.S.	N.S.	N.S.	N.S.	N.S.
Catitti et al., 2022 [43]	Fluorescence activated cell sorting	N.S.	N.S.	N.S.	N.S.
Iyer et al., 2020 [44]	Ultracentrifugation	Ultra-carbon fiber tube	PBS	N.S.	N.S.
Guo et al., 2017 [45]	Ultrafiltration/purification with 30% sucrose D2O cushion and ultracentrifugation	Ultra-clear^TM^ tube (Beckman Coulter)	PBS	Frozen (−80 °C)	N.S.
Xu et al., 2021 [46]	Ultracentrifugation	N.S.	PBS	N.S.	N.S.
Dong et al., 2022 [47]	ExoEasy Maxi kit (Qiagen)	N.S.	PBS	Frozen (−80 °C)	N.S.
Bagio et al., 2023 [48]	Centrifugation	N.S.	PBS	4 °C to −20 °C	7 days
Nilforoushzadeh et al., 2021 [49]	Ultracentrifugation	N.S.	PBS	Frozen (−80 °C)	N.S.
Tao et al., 2017 [50]	Ultrafiltration/purification with 30% sucrose D2O cushion and ultracentrifugation	Ultra-clear^TM^ tube (Beckman Coulter)	PBS	Frozen (−80 °C)	N.S.

**EDTA:** ethylenediamine tetraacetic acid; **NS**: not specified; **PBS:** phosphate buffered saline; **PGE1:** prostaglandin e1.

**Table 4 ijms-24-13043-t004:** Analysis of different markers in PRP-EVs. Markers from the corresponding categories were included (✓) or not (✕) in the articles.

Article	Markers			Size
	Cat. 1 (Transmembrane or GPI-Anchored Proteins)	Cat. 2 (Cytosolic Proteins)	Cat. 3 (Purity Control)	
Liu et al., 2019 [31]	✓ (CD9, CD63, CD81)	✓ (TSG101)	✕	145.6 ± 50.4 nm
Otahal et al., 2020 [32]	✓ (CD14, CD41, CD9, CD63, CD235a)	✓ (ALIX)	✓ (APOA1, APOB100)	***
Otahal et al., 2021 [33]	✓ (CD9, CD63)	✓ (ALIX)	✓ (APOA1, APOB100/48)	***
Zhang et al., 2022 [34]	✓ (CD9, CD63, CD81)	✕	✕	Ranging from 40 to 140 nm
Rui et al., 2021 [35]	✓ (CD9, CD41, CD63, CD81)	✓ (Flotillin, TSG101)	✓ * (Calnexin)	Ranging from 30 to 150 nm
Saumell-Esnaola et al., 2022 [36]	✓ (CD9, CD63, CD81, LAMP1)	✓ (β-actin, Caveolin-1, Flotillin, HSP70)	✓ (APOA1)	Ranging from 20 to 40 nm
Stoner et al., 2016 [37]	✓ (CD61)	✕	✕	***
Yuana et al., 2013 [38]	✕	✕	✕	Ranging from 25 to 260 nm
Zhang et al., 2019 [39]	✓ (CD9, CD61, CD63, CD81)	✕	✕	N.S.
Zhang et al., 2020 [40]	✓ (CD9, CD61, CD63, CD81)	✓ (TSG101)	✕	Peak sizes of 84 nm and 95 nm
Dai et al., 2023 [41]	✓ (CD9, CD41, CD63)	✕	✕	195.6 nm
Xu et al., 2021 [42]	✓ (CD81)	✓ (TSG101)	✕	N.S.
Catitti et al., 2022 [43]	✓ **	✓ **	✓ **	N.S.
Iyer et al., 2020 [44]	✓ (CD9, CD63, CD81)	✕	✕	Ranging from 30 to 200 nm
Guo et al., 2017 [45]	✓ (CD9, CD41, CD63, CD81)	✕	✕	Ranging from 40 to 100 nm
Xu et al., 2021 [46]	✓ (CD9)	✓ (TSG101)	✕	Ranging from 40 to 100 nm
Dong et al., 2022 [47]	✓ (CD9, CD63, CD81)	✓ (ALIX)	✕	145.6 ± 50.4 nm
Bagio et al., 2023 [48]	✓ (CD63, CD81)	✕	✕	Ranging from 44 to 127 nm
Nilforoushzadeh et al., 2021 [49]	✓ (CD9, CD63, CD81)	✓ (TSG101)	✓ * (Calnexin)	Ranging from 50 to 150 nm
Tao et al., 2017 [50]	✓ (CD9, CD41, CD63, CD81)	✓ (TSG101)	✓ * (Calnexin)	Ranging from 30 to 100 nm

***** According to MISEV2018 [18], calnexin is considered as category 4 (proteins localized in membranous intracellular compartments and not enriched in the smaller EVs, endoplasmic reticulum in this case). ****** Too extensive list to be specified in this table; check the article directly. *** Although the graphs are provided, the exact data or range are not specified in the text.

## Data Availability

All data generated in this research was reported in the manuscript.

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
