# Peer review of "Advances in Platelet Rich Plasma-Derived Extracellular Vesicles for Regenerative Medicine: A Systematic-Narrative Review"

_ijms, 2023, doi:10.3390/ijms241713043_

Round 1
Reviewer 1 Report
Manuscript is generally very good This is a very important topic to be discussed. I understand this is a new area of study, however further discussion about the comparison of PRP and EVs released by platelets ( i.e. difference of platelet release when it is activated versus release of EVs when stimulated), how EV can play a role on platelet effects, etc could benefit the paper. I understand this was discussed briefly in the manuscript, but further detail about those differences ( if information is available) would be interesting.
Another important comment is about the figures. Many of the figures need adjustment. A circular graph does not seem to be the most appropriate for the data being presented in many cases ( suggestions and further comments on manuscript attached). The tables are useful and generally good, but the first table needs some editing as well ( more detail on the manuscript attached).
I do think that in general this manuscript is well-written and discuss a very important and current topic in the field.

Reviewer 2 Report
The article is well written, very interest topic for medicine.
Author Response
No questions to be addressed.
Reviewer 3 Report
The Review by Anitua and coworkers covers properly the topic reported in the title. The 'stratergy' for the resarch of the different articles is clearly explained and the results obtained reflect the quality of the approach adopted for the collection of the data. Tables and figures are clear, alsofor readers who are not expert in the field. I recomand the pubblication of the review after a revision of the text for the removal of minor mistakes.
The main question of the review is the evaluation, according to the literature , of the role of PRP derived extracellular vescicles in tissue regenerative processes. This is an original topic, worth to be evaluated. The review by Aniuta and colleagues, that are experts in the field, aids to sched light and fix ‘milestones’ about the articles present in the literature on this specific argument. Moreover the authors propose a methodology that can be translated to a large variety of different topics of biological sciences. The conclusions drawn are consistent with the methodology adopted and moreover, they are clearly divided into different chapters (from 3.6 to 3.11). The references, whose evaluation is focus of the investigation, are appropriate.
The quality of the English language is good, except for minor mistakes that should be corrected
Author Response
No questions to be addressed.